# IMAST: IMPORTANCE-AWARE STATISTICAL TEST FOR TRANSFORMER INTERPRETABILITY EVALUATION

## ABSTRACT

Post-hoc explanations offer a promising avenue to interpret Transformer models. Despite plausible visualizations, rigorous evaluations of their efficacy remain largely unexplored. In this paper, we focus on the principle of faithfulness, a fundamental property of explanation methods: *the importance scores derived from explanation methods should reflect the anticipated impact of corresponding input elements.* To this end, we propose a novel evaluation framework, the **IM**portance-**A**ware **S**tatistical **T**est (**IMAST**). Unlike traditional metrics that rely on cumulative perturbation and quantify performance reduction, IMAST performs statistical comparisons among individual pixel subsets, thereby aggregating their importance score differences into a resulting faithfulness coefficient. Extensive experiments demonstrate the shortcomings of existing metrics in aligning with the faithfulness assumption, as they often cannot distinguish Random Attribution from advanced explanations. In contrast, IMAST is effective in setting a baseline for evaluating faithfulness, which provides a robust benchmark for explanations. Moreover, using the proposed IMAST, we find through ablation studies that the incorporation of gradient information and cross-layer aggregation significantly improves the faithfulness of attention-based methods, providing guidance for the future development of Transformer interpretability.

## 1 INTRODUCTION

The extensive application of Transformers in computer vision has emphasized the need to unravel their inherent black-box characteristics (Vaswani et al., 2017; Carion et al., 2020; Dosovitskiy et al., 2021). This challenges traditional post-hoc explanation methods which are originally designed for MLPs and CNNs. Consequently, an emerging line of work focuses on developing new paradigms specific to Transformers, where attention mechanisms play a dominant role (Abnar & Zuidema, 2020; Chefer et al., 2021b;a; Qiang et al., 2022; Ali et al., 2022). Integrating the attention distribution, these explanation methods can produce visually convincing heatmaps that resonate with human intuition (Colin et al., 2022). However, recent works (Jacovi & Goldberg, 2020; DeYoung et al., 2020) claimed that it is crucial to evaluate how accurately these interpretations reflect the true reasoning process of the Transformer model and termed this aspect as *faithfulness*.

To elucidate a model's predictions, post-hoc explanations attribute each input element with a value signifying its contribution to the model. This value, thus assigned, is referred to as an *importance score*. Recent studies on evaluating the explanations commonly adopt an ablation approach. This involves perturbing input features, such as image pixels, that are identified as most or least important by the explanation method under examination. For example, they perturb pixels with the highest importance scores and then observe whether there is a decrease in the predictive quality, which serves as a surrogate examination (Shrikumar et al., 2017a; Nguyen, 2018; Atanasova et al., 2020; Chen et al., 2020; Liu et al., 2022). Despite the prevalence of these strategies, our study reveals that they all overlook a proper evaluation of the faithfulness of explanations. We advance our discussion by characterizing the core assumption of faithfulness that underpins explanation methods:

**Core Assumption.** *The magnitude of importance scores signifies the level of anticipated impacts. Consequently, (i) input elements assigned higher scores are expected to exert greater influence on the model's prediction, compared with those with lower scores, and (ii) two elements with larger differences in their importance scores are expected to cause a greater disparity in their influences.*

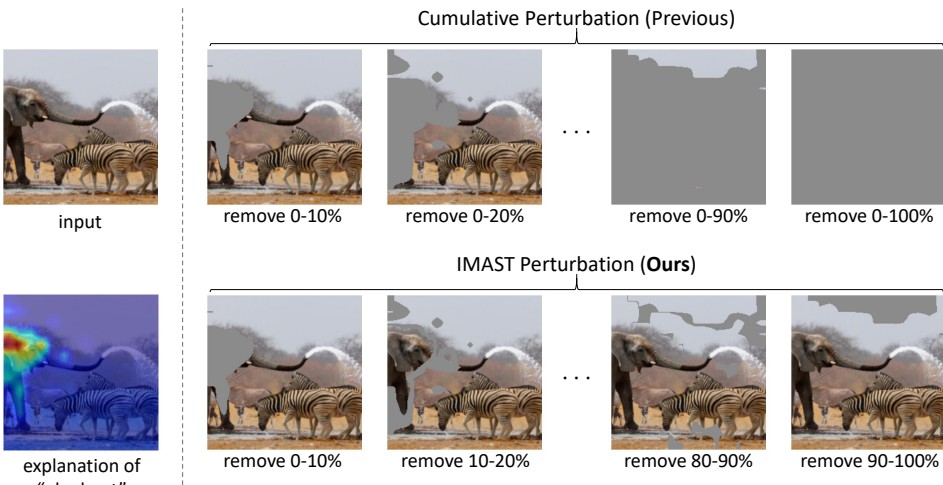

Figure 1: Explanation result and illustration of two different perturbation manners: cumulative perturbation and our IMAST perturbation. Previous metrics perturb the pixel subsets cumulatively. In contrast, the IMAST perturbs them individually to directly compare their influences.

In response to these insights, for a thorough faithfulness evaluation, it is necessary to: **(i)** explicitly compare the influences of input elements with different levels of importance, and **(ii)** quantify the differences in importance scores to reflect the expected disparities in their impacts. However, existing metrics fall short in both aspects, as they rely on cumulative perturbation (Shah et al., 2021) and do not consider the information embedded in the magnitude of importance scores. For example, with cumulative perturbation (see Figure 1), it is convoluted to discern the influence of pixels ranked between the top 0-10% (the elephant's body) and 90-100% (the sky) of importance. This is due to the removal of the top 90-100% important pixels only after the top 0-90% have been eliminated, which conflates their individual impacts. Moreover, without considering the exact values of importance, it is uncertain to what degree an explanation expects pixels in the top 0-10% to have more influence than those in 90-100%. Existing metrics cannot adequately evaluate an explanation's ability to differentiate importance levels among different pixels, thereby failing to validate the core assumption. Such deficiency can lead to unreliable outcomes, underlining the need for a nuanced evaluation. For instance, it is alarming that commonly used metrics fail to distinguish between some SOTA methods and Random Attribution (Wang & Wang, 2022).

Recognizing that faithfulness is essential for explanation methods to depict models' behavior, we propose a novel evaluation framework, **IM**portance-**A**ware **S**tatistical **T**est, or **IMAST**. This framework analyzes how well an explanation method aligns with faithfulness. The proposed IMAST operates by conducting a statistical analysis of pixel subsets with varying importance scores and comparing their impacts on the model's prediction. The importance score distribution is evaluated based on its alignment with the true effect of corresponding input elements. For instance, if a pixel subset with higher importance scores significantly impacts the model's prediction more than a subset with lower scores, as anticipated, such a pair of subsets is deemed to satisfy the faithfulness criterion. Consequently, the disparity in importance scores between these two subsets, which represents the degree of expectation, will be positively accumulated to the test outcome. Conversely, if a pair of subsets does not meet this expectation, it is identified as a violator and will have a negative contribution. The IMAST is suitable for testing the core assumption validity, as it involves explicit comparisons among different elements and captures the expected disparities in their impacts.

Experimental results across a range of datasets and models in Section 5 reveal that current metrics ignore proper evaluations of faithfulness. Furthermore, we observe that most explanation methods actually underperform when tested against the core assumption. To investigate the key factors that affect faithfulness, we perform ablative experiments on attention-based explanation methods.

In summary, we state our contributions as follows: **(i)** We develop a new evaluation framework, IMAST, to assess how well explanations adhere to the core assumption of faithfulness. By examining ten representative explanation methods across three datasets and three Transformer models, we demonstrate that IMAST can provide a complementary tool for examining how importance scores

signify the level of anticipated impacts. **(ii)** Empirically, we demonstrate IMAST's capability to distinguish meaningful explanations from Random Attribution, setting a useful and robust benchmark. **(iii)** We provide insights into certain designs in current explanation methods that may alter faithfulness. Our ablative results highlight the role of gradient information and aggregation rules, guiding the path for future improvements in Transformer interpretability methodology.

## 2 RELATED WORK

### 2.1 POST-HOC EXPLANATIONS

**Traditional post-hoc methods.** Post-hoc explanation methods largely fall into two groups: gradient-based and attribution-based approaches. The first group includes methods like Input $\odot$ Gradient (Shrikumar et al., 2017b), SmoothGrad (Smilkov et al., 2017), Full Grad (Srinivas & Fleuret, 2019), Integrated Gradients (Sundararajan et al., 2017), and Grad-CAM (Selvaraju et al., 2017). These methods leverage gradient information to calculate importance scores. In contrast, attribution-based methods (Bach et al., 2015; Shrikumar et al., 2017a; Nam et al., 2020; Gur et al., 2021) propagate classification scores backward to the input and then use the resultant values as indicators of contribution. Beyond these two types, there are also other methods, such as saliency-based (Zhou et al., 2016; Mahendran & Vedaldi, 2016; Zhou et al., 2018), Shapley additive explanation (Lundberg & Lee, 2017), and perturbation-based methods (Fong & Vedaldi, 2017; Fong et al., 2019). Although initially designed for MLPs and CNNs, some of them have been successfully adapted for Transformers in recent works (Chefer et al., 2021b; Ali et al., 2022).

**Leveraging attentions for interpreting Transformers.** A distinct branch of research in interpretability is committed to creating new paradigms specifically for Transformers. Attention maps are widely used in methodologies associated with this direction, as they inherently constitute distributions that represent the sampling weights over input features. Representative methods include Transformer Attribution (Chefer et al., 2021b) and Conservative LRP (Ali et al., 2022), two variants of Layer-wise Relevance Propagation tailored for the self-attention mechanism, Transformer-MM (Chefer et al., 2021a), a general explanation framework utilizing gradient information, and ATTCAT (Qiang et al., 2022), a method that formulates Attentive Class Activation Tokens to estimate the relative importance among input elements.

### 2.2 EVALUATION OF FAITHFULNESS

In this paper, we evaluate the faithfulness of explanation methods. This is a critical task (Adebayo et al., 2018), especially given the recent debates concerning whether parameters and features in Transformers are explainable. For instance, the reliability of attention has been questioned in several studies (Jain & Wallace, 2019; Serrano & Smith, 2019), premised on the hypothesis that attention weights should not conflict with gradient rankings. Following this, Wiegreffe & Pinter (2019) proposed alternative testing strategies and argued that attention is interpretable when limited to certain circumstances. However, we find that current studies tend to neglect the values of importance scores and the model's confidence in its original predictions, which deviates from the core assumption of faithfulness and leads to inconsistent and untrustworthy outcomes.

Evaluating post-hoc explanations poses a challenge in the realm of interpretability. Given the absence of justified ground truth (Agarwal et al., 2022), one stream of research focuses on developing human-centered evaluations (Lage et al., 2018; Ross & Doshi-Velez, 2018; Colin et al., 2022). These studies scrutinize the practical value of explanations provided to the end user. In another direction, Adebayo et al. (2018) employed a series of sanity checks to assess changes in explanations *w.r.t.* randomization within the model and dataset. Unlike these studies, our work is broadly related to faithfulness metrics that evaluate explanations by monitoring the model's performance on perturbed data (Atanasova et al., 2020). Various versions of these perturbation-based metrics have been introduced, which provide measures of input feature impact (Shrikumar et al., 2017a; Nguyen, 2018; Chen et al., 2020; DeYoung et al., 2020). Despite the achievements, these approaches use cumulative perturbation without directly contrasting input features of different importance levels, and overlook the values of importance scores in their evaluations. This oversight contributes to deficiencies present in existing methods. Our approach, on the other hand, scrutinizes the model's response to each distinct subset of the input and sheds light on the relevance of the importance scores.

## 3 IMPORTANCE-AWARE STATISTICAL TEST (IMAST)

For image classification, an input image comprises $HW$ elements (pixels in RGB). Given an input image $\mathbf{x}$ and the corresponding predicted class $\hat{y}(\mathbf{x}) \in \{1, 2, ..., C\}$, where $C$ is the number of classes under consideration, the post-hoc explanation generates an importance map $\mathbf{M}(\mathbf{x}, \hat{y}) \in \mathbb{R}^{HW}$. The value of each entry in $\mathbf{M}(\mathbf{x}, \hat{y})$ ought to reflect the contribution of the corresponding element. However, the reliability of these interpretation results remains an open problem. This underscores the necessity for further examination of faithfulness.

Following the core assumption, the property being investigated is the *extent* to which these importance scores are faithful to the model's actual behavior. Therefore, our proposed evaluation is designed to assess how effectively the disparity in importance scores signifies the variation in the influence on the model's confidence. Considering a sample $\mathbf{x}$, we reorder the input elements based on their estimated importance and partition them into $K$ equally sized subsets: $G_1, G_2, ..., G_K$. Each subset $G_i$ comprises pixels with top importance ranking from $(i-1)\frac{HW}{K}$ to $i\frac{HW}{K}$ (Nguyen, 2018; Chen et al., 2020). Regarding each $G_i$ as a basic unit, we define the importance of a subset:

$$s(G_i) = \sum_{p \in G_i} \mathbf{M}(\mathbf{x}, \hat{y})_p, \quad \text{where} \quad i = 1, 2, ..., K. \tag{1}$$

In essence, the importance of a subset $G_i$ is the sum of importance scores over all pixels in $G_i$. Following the convention in literature, we adopt a proxy measure to access the model's behavior: we replace pixels that belong to a certain subset with the per-sample mean value (Hooker et al., 2019) and then observe the resulting effect on the model's confidence. Formally, we represent the replacement result by $Rp(\mathbf{x}, G_i)$. Therefore, the alterations in the model's prediction can be denoted as follows:

$$\nabla pred(\mathbf{x}, G_i) = p(\hat{y}(\mathbf{x})|\mathbf{x}) - p(\hat{y}(\mathbf{x})|Rp(\mathbf{x}, G_i)). \tag{2}$$

The fundamental principle underpinning the IMAST is that a subset $G_i$ of higher importance should exert more effects compared to a subset $G_j$ of significantly lower importance. Specifically, if $s(G_i) \geq s(G_j)$, we expect the following inequality to be upheld:

$$\nabla pred(\mathbf{x}, G_i) \geq \nabla pred(\mathbf{x}, G_j). \tag{3}$$

As the difference between $s(G_i)$ and $s(G_j)$ expands, our expectation for Inequality (3) to hold will intensify. Following this, the growing difference in importance scores should accentuate its influence on the evaluation result. For example, a violation of Inequality (3) should be penalized more when the difference in importance becomes larger, thus better reflecting the deviation from the expected model behavior.

Inspired by the Kendall $\tau$ statistic (Kendall, 1938), we look into all possible pairs of $G_i$ and $G_j$ and assess the compliance with Inequality (3). When this inequality is violated, the difference in importance will negatively impact the evaluation result. On the contrary, when the inequality holds true, the importance difference will add positively to the evaluation. For example, suppose for a pair of subsets $G_i$ and $G_j$ with $s(G_i) \geq s(G_j)$, we find $\nabla pred(\mathbf{x}, G_i) < \nabla pred(\mathbf{x}, G_j)$. Then, the difference in importance, $s(G_i) - s(G_j)$, is considered a penalty that reflects the magnitude of our unfulfilled expectations and will be subtracted from the overall result. If we observe $\nabla pred(\mathbf{x}, G_i) \geq \nabla pred(\mathbf{x}, G_j)$ as expected, the difference $s(G_i) - s(G_j)$ will serve as a reward and positively contribute to the evaluation outcome. Detailed steps are elaborated in Algorithm 1.

As per its definition, the IMAST produces a faithfulness coefficient, denoted as $F$, that ranges from $[-1, 1]$. The sign of $F$ reveals the direction of correlation, *i.e.*, it evaluates if the input elements with higher importance scores generally exhibit greater or lesser predictive influence on the model. Beyond just the direction, the absolute value of $F$ quantitatively measures the degree of correlation.

## 4 EXPERIMENTAL SETUP

### 4.1 DATASETS AND MODELS

We utilize three benchmark image datasets: CIFAR-10, CIFAR-100 (Krizhevsky et al., 2009), and ImageNet (ILSVRC) 2012 (Russakovsky et al., 2015). Details regarding the scales of data, numbers of classes, and image resolutions for each dataset are provided in Appendix A.1. Furthermore, to

---

**Algorithm 1** Importance-Aware Statistical Test (IMAST)

---

1: **Input:** Pre-trained model $\Phi$, explanation method $\mathcal{E}$, input image $\mathbf{x}$.
2: **Output:** Faithfulness coefficient $F$.
3: **Initialization:** $F \leftarrow 0, totalWeight \leftarrow 0$
4: Compute the importance map $\mathbf{M}(\mathbf{x}, \hat{y})$ based on $\Phi$, $\mathcal{E}$, and $\mathbf{x}$. Generate $G_i$ and obtain corresponding $s(G_i)$ and $\nabla pred(\mathbf{x}, G_i)$, for $i = 1, 2, ..., K$.
5: **for** $i = 1$ to $K - 1$ **do**
6:     **for** $j = i + 1$ to $K$ **do**
7:         **if** $\nabla pred(\mathbf{x}, G_i) \geq \nabla pred(\mathbf{x}, G_j)$ **then**
8:             $weight \leftarrow s(G_i) - s(G_j)$
9:         **else**
10:             $weight \leftarrow -(s(G_i) - s(G_j))$
11:         **end if**
12:         $F \leftarrow F + weight$
13:         $totalWeight \leftarrow totalWeight + |weight|$
14:     **end for**
15: **end for**
16: $F \leftarrow F/totalWeight$
17: **Return** $F$

---

ensure the reliability of our evaluation, we experiment with three state-of-the-art models that are widely adopted in this field: ViT-B, ViT-L (Dosovitskiy et al., 2021), and DeiT-B (Touvron et al., 2021). In these models, images are divided into non-overlapping $16 \times 16$ patches, then flattened and processed to create a vector sequence. Similar to BERT (Devlin et al., 2019), a classification token $[CLS]$ is prepended to the beginning of the sequence for classification purposes.

## 4.2 EXPLANATION METHODS

We investigate ten explanation methods spanning three categories, *i.e.*, gradient-based, attribution-based, and attention-based. Each method holds unique assumptions about the network architecture and information propagation. For a better assessment, selected methods are widely recognized in the explainability literature and also compatible with the Transformer models under consideration. Detailed descriptions of these techniques are in Appendix A.2.

**Gradient-based methods.** This group of approaches includes Integrated Gradients (Sundararajan et al., 2017) and Grad-CAM (Selvaraju et al., 2017). Note that Integrated Gradients is model-agnostic, namely, only the gradient and the input are required. Thus, it can be applied to Vision Transformers without modifications. As for Grad-CAM which is initially designed for CNNs, our implementation follows the prior study in Transformer interpretability (Chefer et al., 2021b).

**Attribution-based methods.** Unlike the gradient-based methods, attribution-based methods explicitly model the information flow inside the network. We select LRP (Binder et al., 2016), Partial LRP (Voita et al., 2019), Conservative LRP (Ali et al., 2022), and Transformer Attribution (Chefer et al., 2021b)) in our experiment for a thorough analysis.

**Attention-based methods.** Regarding the attention-based methods, we employ four variants: Raw Attention, Rollout (Abnar & Zuidema, 2020), Transformer-MM (Chefer et al., 2021a), and ATTCAT (Qiang et al., 2022) in our experiments. These methods are specifically designed for Transformers.

## 4.3 EVALUATION METRICS

We compare our proposed IMAST with widely adopted existing metrics to validate its reliability.

**Area Under the Curve (AUC)** ↓**.** This metric calculates the Area Under the Curve (AUC) corresponding to the model's performance as different proportions of input features are perturbed (Atanasova et al., 2020). To elaborate, we first generate new data by gradually removing features in increments of 10% (from 0% to 100%) based on their estimated importance weights. The model's accuracy is then assessed on these perturbed data, resulting in a sequence of accuracy measurements. The AUC is subsequently computed using this sequence.

**Area Over the Perturbation Curve (AOPC) ↑.** Rather than measuring the model's accuracy, AOPC (Nguyen, 2018; Chen et al., 2020) quantifies the variations in output probabilities *w.r.t.* the predicted label after perturbations. A higher AOPC indicates a better explanation.

**Log-odds score (LOdds) ↓.** The LOdds (Shrikumar et al., 2017a; Qiang et al., 2022) evaluates if the input elements, considered as important, are enough to sustain the model's prediction, and this is measured on a logarithmic scale. To facilitate fair and reliable comparisons, we gradually eliminate the top 0%, 10%, ..., 90%, and 100% of features, based on their importance. This approach aligns with the methodologies employed for calculating AUC and AOPC.

**Comprehensiveness (Comp.) ↓.** The Comprehensiveness (DeYoung et al., 2020) measures if input elements with lower importance are dispensable for the model's prediction. For consistent comparisons, we cumulatively eliminate features in the least important 0%, 10%, ..., 90%, and 100%.

## 5 EXPERIMENTAL RESULTS

### 5.1 ANALYZING INTERRELATIONSHIPS AMONG EVALUATION METRICS

To demonstrate the significance and necessity of IMAST, we conduct a correlation analysis following the thorough experimental setup. To this end, we begin by evaluating each explanation method on single samples independently, using each of the metrics. Then, we compute the rank correlations between the evaluation results obtained from the IMAST and those from existing metrics. Note that we are correlating based on the rankings rather than the exact values of evaluation results, as these metrics can vary in scales and orientations. This approach allows us to quantify the degree of similarity between the assessments provided by the IMAST and the traditional metrics in current use. Figure 2 shows the

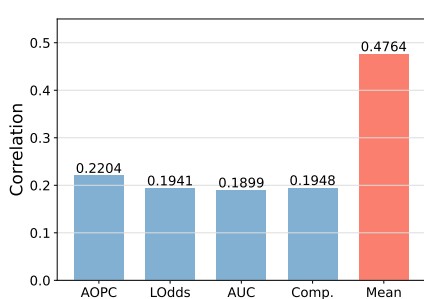

Figure 2: Correlations between sample rankings *w.r.t.* our IMAST and existing metrics.

comprehensive correlation results averaged across all considered datasets, explanation methods, and models. The first four bars on the left depict the correlations between our IMAST and the metric indicated by the corresponding x-axis label. The rightmost bar shows the average correlation among all other metrics, excluding ours.

As displayed, the correlation scores between our IMAST and other existing metrics range from 0.18 to 0.22 (in this analysis, a result of 1 indicates a complete correlation, while a result of 0 indicates no correlation). These low scores signify minimal congruence in their evaluation, indicating that existing metrics may not effectively capture the adherence of an explanation to the central assumption of faithfulness. In essence, the traditional metrics tend to generate similar results regardless of the degree of faithfulness, due to their lack of comparisons among individual input elements and the consideration of differences in importance scores. On the other hand, the average intra-correlation among the existing metrics themselves is noticeably higher. This implies that they actually evaluate overlapping aspects of interpretations (mainly the effect of progressive feature removal) while overlooking the faithfulness assumption. These findings emphasize the necessity of our IMAST, as current metrics appear to lack the capabilities to adequately assess faithfulness, reinforcing the need for a more comprehensive evaluation for explanations.

### 5.2 EVALUATING RANDOM ATTRIBUTION AS AN "EXPLANATION"

We now turn our attention to a critical examination: evaluating the performance of Random Attribution as explanation results (Hooker et al., 2019; Shah et al., 2021; Wang & Wang, 2022). In this context, Random Attribution assigns importance scores to input elements in a purely stochastic manner, with no reference to the model's internal operations or inference process. For example, one can generate an importance heatmap using a uniform distribution. From the perspective of the core assumption, such Random Attribution represents a complete absence of faithfulness, as the assigned importance scores bear no relationship to the actual impact of each input element. As a result, an

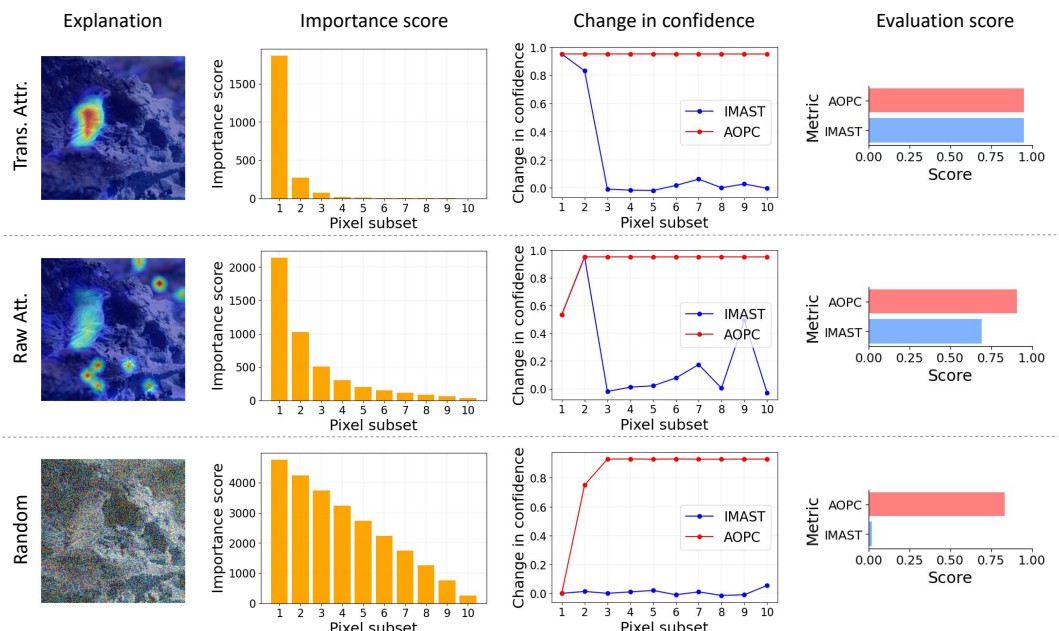

Figure 3: Illustration of three explanations for the predicted class 'linnet', importance score distributions, changes in confidence caused by perturbation, and final IMAST and AOPC scores.

ideal metric for assessing faithfulness should return a distinct score when applied to Random Attribution, essentially setting a baseline indicative of a complete lack of meaningful understanding of the model's prediction. In particular, from the viewpoint of mathematical expectation, our IMAST yields a correlation coefficient of zero for Random Attribution. Given that the importance scores are sampled from a random distribution, the subsets $(G_i, i = 1, 2, ..., K)$ partitioned according to these scores will not show significant differences in their discriminative information. Therefore, Inequality (3) holds true with a probability of one-half, leading to a balance between the positive and negative terms during the calculation outlined in Algorithm 1.

### 5.2.1 A CASE STUDY

We first present a case study by comparing IMAST's behavior with AOPC. AOPC operates on cumulative perturbation and disregards the alignment between importance values and actual influences.

As shown in Figure 3, we evaluate three explanation methods: Transformer Attribution (Chefer et al., 2021b), Raw Attention, and Random Attribution. Specifically, following the literature convention, we divide the image into ten disjoint pixel subsets, designated by $K=10$. The importance score of each subset $s(G_i)$ is computed using Equation 1. We then implement both IMAST's perturbation and AOPC's cumulative perturbation, and calculate the resulting changes in the model's confidence (probability) for the predicted class, as defined by Equation 2. Here, a positive change signals a decrease in confidence. For all three methods, the confidence drops induced by cumulative perturbation remain almost unchanged (above 0.9) after the removal of the top 20% important pixels, despite additional pixels being progressively removed. This results in consistently high AOPC scores. As previously discussed in Section 1, this pattern emerges because less important pixels are removed in sequence only after the most important pixels have been eliminated, causing their individual effects to become entwined. Conversely, IMAST individually assesses the influences of pixel subsets and measures their alignment with the importance distribution. In the case of Transformer Attribution, the influences of the subsets (marked by the blue curve) closely align with the importance score distribution, yielding a high IMAST score. For Raw Attention, subsets $G_i(i = 3, 4, 5, 6)$ have minimal influences, likely due to unexpected emphasis on irrelevant objects and background features, leading to a lower IMAST score. Lastly, for Random Attribution, the influences of subsets show little variation, resulting in a near-zero IMAST score. This case study shows that IMAST provides a rigorous evaluation that effectively differentiates superior and inferior explanations.

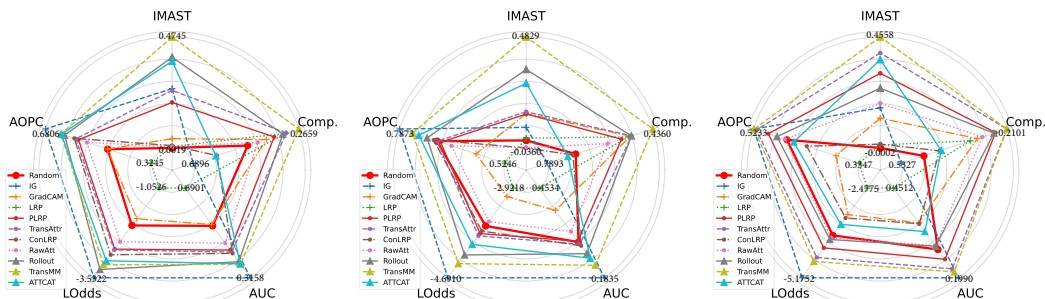

Figure 4: Evaluation results for explanation methods and Random Attribution (red). Three graphs present results on CIFAR-10 (left), CIFAR-100 (middle), and ImageNet (right), respectively. The values on each axis have been rescaled so that a larger distance from the center consistently signifies superior performance. Appendix D presents enlarged graphs for better clarity.

### 5.2.2 LARGE-SCALE EXPERIMENTS

To further validate the efficacy of IMAST, we conduct experiments on large-scale datasets. Figure 4 presents the overall results. Across all datasets, IMAST consistently scores Random Attribution near zero, while most explanation methods obtain positive scores under IMAST, demonstrating its ability to set a standard benchmark. In contrast, other metrics cannot provide consistent evaluation for Random Attribution. For example, under these metrics, Random Attribution appears to fare better than some state-of-the-art methods introduced in recent years, such as Partial LRP (Voita et al., 2019), Transformer Attribution (Chefer et al., 2021b), and ATTCAT (Qiang et al., 2022) (see Figure 4). Another observation is that existing metrics seem to be sensitive to the removal order of the cumulative perturbation they employ (Rong et al., 2022). A phenomenon across three datasets in our experiment exemplifies this problem: while I.G. performs best on AOPC, LOdds, and AUC (with Most Relevant First removal), it is surprisingly the worst on Comprehensiveness (with the reverse removal order). This indicates that current metrics are significantly inconsistent *w.r.t.* hyperparameters such as removal orders (Rong et al., 2022). In contrast, we eliminate these inconsistencies by directly comparing individual subsets instead of cumulatively removing them in a specific order.

### 5.3 ASSESSMENT OF CURRENT EXPLANATION METHODS BASED ON IMAST

Figure 4 illustrates the results of IMAST for explanation methods over all utilized models and datasets. Observations from our results indicate that all explanation methods perform moderately, as reflected in their suboptimal IMAST scores. This suggests the need for potential improvements in better portraying the model's prediction process and aligning with the core faithfulness assumption. Furthermore, of all methods evaluated, we can see that those utilizing attention mechanisms generally perform best. However, despite falling under the same category, Raw Attention noticeably underperforms. This leads us to hypothesize that attention-based methods can only achieve superior performance when they incorporate auxiliary information, such as gradient and cross-layer integration. We further conduct experiments in Section 5.4 for its demonstration.

### 5.4 EFFECTS OF SPECIFIC DESIGNS IN EXPLANATION METHODS

We now delve into the factors that significantly augment the alignment with the faithfulness assumption. We focus our analysis on attention-based explanation methods because attention is inherently meaningful for Transformers. Moreover, previous assessments have demonstrated that attention-based methods generally outperform others, making them a valuable subject of further investigation.

As depicted in Figure 4, it appears that explanation methods utilizing well-crafted aggregation rules and auxiliary information such as gradient tend to score higher under IMAST. Drawing on prior research (Selvaraju et al., 2017; Abnar & Zuidema, 2020; Srinivas & Fleuret, 2021), we hypothesize that the design of aggregation rules and the integration of gradient play a vital role in compliance with faithfulness. To validate this, we conduct ablative experiments employing four variants of attention-based methods: **(i)** utilizing only the final layer of the model, **(ii)** aggregating information

Table 1: Ablation study on attention-based explanation methods.

| w/o aggregation | w/ aggregation | w/o gradient | w/ gradient | IMAST ↑ |
|:---:|:---:|:---:|:---:|:---:|
| ✓ | | ✓ | | 0.1835 |
| | ✓ | ✓ | | 0.2453 |
| ✓ | | | ✓ | 0.3783 |
| | ✓ | | ✓ | **0.4558** |

across all layers, **(iii)** utilizing only the final layer while integrating gradient information, and **(iv)** aggregating information across all layers and also integrating gradient information.

Table 1 presents our ablation study's results on attention-based explanation methods, revealing some insightful patterns. Firstly, the incorporation of gradient information significantly enhances our faithfulness scores. When considering only the last layer, the introduction of gradients boosts the evaluation outcome by approximately 106%. This impact persists even when aggregation is applied across all layers, with a relative increase of about 86%. Secondly, despite being less influential than gradient information, aggregating across multiple layers also positively impacts the outcomes. This effect occurs irrespective of whether the gradient is employed, hinting that a more holistic view of the model can consistently facilitate more faithful explanations. These findings empirically affirm our initial hypothesis that both the gradient and aggregation rules are crucial for Transformer interpretability, with the former playing a more dominant role. By optimizing these factors, one may be able to design more advanced interpretations.

## 5.5 EXPLORING INFLUENTIAL FACTORS IN IMAST

### 5.5.1 MEASURE OF DISTINCT IMPORTANCE SCORES

In our proposed Algorithm 1, we quantify the extent of satisfying or violating our expectation by differences in importance scores, expressed as $weight \leftarrow s(G_i) - s(G_j)$. One possible alternative for this measure is taking the ratio: $weight \leftarrow \frac{s(G_i)}{s(G_j)}$. This seems promising because ratios are effective at capturing the relative importance between input elements. However, using a ratio neglects the property of scale-invariance. In practice, the explanation results may be normalized or scaled to $[0, 1]$ as a post-processing (Selvaraju et al., 2017; Chefer et al., 2021a; Qiang et al., 2022), which will skew the ratio calculations. Additionally, it may cause extremely high or even infinite ratios when $s(G_j)$ is close to zero after transformation. This issue is especially problematic while comparing explanations from different methods that scale their importance scores differently. In contrast, our designed IMAST maintains the scale-invariance property. Regardless of the scale on which importance scores are expressed, the results remain consistent, enabling a stable comparison between distinct methods and thereby ensuring a robust evaluation. Formal proof demonstrating that our IMAST satisfies the scale-invariance property is provided in Appendix C.1.

### 5.5.2 OTHER CHOICES OF K

To explore the sensitivity of IMAST to hyperparameters, we study the effects of different choices of $K$. Detail of more experiments is provided in Appendix B.1. Our analysis shows that while the choice of $K$ impacts the granularity of IMAST, its underlying principles remain consistent. The choice of $K$ indicates a trade-off between computational efficiency and detailed insights.

## 6 CONCLUSION

In this work, we propose IMAST, a new metric of faithfulness. IMAST leverages importance-aware comparisons of pixel subsets that differ in their prediction contribution, which provides a unique and robust perspective. Comprehensive experiments reveal insightful observations: **(i)** Our correlation analysis shows the necessity of the IMAST, as other metrics capture overlapping aspects of interpretations while neglecting faithfulness. **(ii)** Unlike existing metrics, the IMAST can identify Random Attribution as completely lacking significant information and provides consistent results free of removal order dependency. **(iii)** Attention-based methods generally perform better on faithfulness, and their performance can be enhanced significantly by employing gradient information and inter-layer integration. Overall, our work provides a complementary framework for evaluating Transformer explanation methods, which will spur further research in the development of explainable AI.

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

# A EXPERIMENTAL SETUP

## A.1 DATASETS

**CIFAR-10 and CIFAR-100.** CIFAR-10 and CIFAR-100 (Krizhevsky et al., 2009) are two widely used image classification datasets, each containing 60,000 $32 \times 32$ color images. CIFAR-10 has 10 classes, while CIFAR-100 has a more challenging setting with 100 classes. Both datasets are split into 50,000 training and 10,000 testing images.

**ImageNet.** ImageNet dataset (Russakovsky et al., 2015) is a large-scale benchmark in the field of image classification. In this work, we evaluate explanation methods on the ImageNet validation set, which comprises 50,000 high-resolution images across 1,000 distinct classes. Each class contains roughly the same number of images, ensuring a balanced benchmark.

## A.2 EXPLANATION METHODS

### A.2.1 GRADIENT-BASED METHODS

**Integrated Gradients.** Integrated Gradients (Sundararajan et al., 2017) calculates contributions by integrating gradients along a path from a baseline input $\mathbf{x_0}$ to the original input $\mathbf{x}$:

$$\text{Integrated Gradients}(\mathbf{x}, \mathbf{x_0}) = (\mathbf{x} - \mathbf{x_0}) \odot \int_0^1 \frac{\partial f(\mathbf{x_0} + \alpha(\mathbf{x} - \mathbf{x_0}))}{\partial \mathbf{x}} \mathrm{d}\alpha, \tag{4}$$

where $f$ represents the classification model. In practice, the integral is approximated using the Riemann Sum over a linear interpolation path.

**Grad-CAM.** Instead of the original input, Grad-CAM (Selvaraju et al., 2017) utilizes the attention map in the last layer. Following the prior work (Chefer et al., 2021b), we perform multi-head integration based on gradient information.

A.2.2 ATTRIBUTION-BASED METHODS

**LRP.** LRP (Binder et al., 2016) starts from the model's output and propagates relevance scores backward up to the input image. This propagation adheres to a set of rules defined by the Deep Taylor Decomposition theory (Montavon et al., 2017).

**Partial LRP.** Partial LRP (Voita et al., 2019) also backpropagates relevance scores, but uniquely, it uses the relevance map from a specific intermediate layer as the final explanation. In line with convention, we choose the relevance map associated with the attention map in the last layer.

**Transformer Attribution.** Transformer Attribution (Chefer et al., 2021b) is an attribution method specifically designed for Transformer models. It first computes relevance scores via the LRP, and then integrates these scores with attention maps to produce an explanation.

**Conservative LRP.** Conservative LRP (Ali et al., 2022) introduces specialized Layer-wise Relevance Propagation rules for attention heads and layer norms in Transformer models. This is designed to implement conservation, a common property of attribution techniques.

A.2.3 ATTENTION-BASED METHODS

**Raw Attention.** This method (Jain & Wallace, 2019) extracts the multi-head attention map from the last layer of the model and reshapes the row corresponding to the $[CLS]$ token into the 2D input space. An interpretation is further derived by averaging across different heads.

**Rollout.** Rollout (Abnar & Zuidema, 2020) interprets the information flow within Transformers from the perspective of Directed Acyclic Graphs (DAGs). It traces and accumulates the attention weights across various layers using a linear combination strategy.

**Transformer-MM.** Transformer-MM (Chefer et al., 2021a) is a general interpretation framework applicable to diverse Transformer architectures. It aggregates attention maps with corresponding gradients to generate class-specific explanations.

**ATTCAT.** ATTCAT (Qiang et al., 2022) is a Transformer explanation technique using attentive class activation tokens. It employs a combination of encoded features, their associated gradients, and their attention weights to produce confident explanations.

A.3 EVALUATION METRICS

**Area Under the Curve (AUC)** ↓**.** The definition of the AUC metric (Atanasova et al., 2020) can be found in Section 4.3. This metric measures the performance of a model across various levels of feature perturbation.

**Area Over the Perturbation Curve (AOPC)** ↑**.** AOPC (Nguyen, 2018; Chen et al., 2020) measures the changes in output probabilities *w.r.t.* the predicted label after perturbations:

$$\text{AOPC} = \frac{1}{|K|} \sum_{k \in K} (\hat{p}(y|\mathbf{x}) - \hat{p}(y|\mathbf{x_k})), \tag{5}$$

where $K = \{0, 10, ..., 90, 100\}$ is a set of perturbation levels, $\hat{p}(y|\mathbf{x})$ estimates the probability for the predicted class given a sample $\mathbf{x}$, and $\mathbf{x_k}$ is the perturbed version of $\mathbf{x}$, from which the top $k\%$ elements ranked by relevance scores are eliminated.

**Log-odds score (LOdds)** ↓**.** LOdds (Shrikumar et al., 2017a; Qiang et al., 2022) averages the difference between negative logarithmic probabilities on the predicted label before and after masking $k\%$ top-scored elements over perturbations $K$:

$$\text{LOdds} = -\frac{1}{|K|} \sum_{k \in K} \log \frac{\hat{p}(y|\mathbf{x})}{\hat{p}(y|\mathbf{x_k})}. \tag{6}$$

The notations are the same as in Equation 5.

**Comprehensiveness (Comp.)** ↓**.** Comprehensiveness (DeYoung et al., 2020) is also referred to as the negative perturbation test. This examines how the removal of supposedly less important input elements affects the model's output. Concretely, Comprehensiveness gauges the shifts in output probabilities *w.r.t.* the predicted label after the least important features have been excluded.

Table 2: Performances of current methods on our IMAST, using different values of $K$. Results are averaged over three Transformer models on ImageNet.

| $K$ | I. G. | Grad-CAM | LRP | P. LRP | Trans. Attr. | Con. LRP | Raw Att. | Rollout | Trans. MM | ATTCAT |
|---|---|---|---|---|---|---|---|---|---|---|
| 5 | 0.1585 | 0.1659 | 0.0246 | 0.4628 | 0.5680 | 0.0544 | 0.3200 | 0.3372 | 0.6041 | 0.3178 |
| 10 | 0.1647 | 0.1142 | 0.0120 | 0.3066 | 0.3902 | 0.0155 | 0.1835 | 0.2453 | 0.4558 | 0.3629 |
| 20 | 0.1785 | 0.1000 | 0.0054 | 0.2282 | 0.2906 | -0.0201 | 0.1411 | 0.1956 | 0.3651 | 0.3617 |

## B  MORE EXPERIMENTAL RESULTS

### B.1  DIFFERENT CHOICES OF K

To explore the sensitivity of the proposed IMAST to the hyperparameter, we conduct experiments with different choices of $K$, specifically $K$=5, $K$=10, and $K$=20. Here, $K$ denotes the number of subsets that the input elements are partitioned into.

Figure 5 presents the correlations between IMAST results with different values of $K$. The positive correlation between $K$=5 and $K$=10, demonstrated by a coefficient of 0.6101, indicates a moderate similarity in IMAST results for these two settings. This suggests that a coarse partitioning into five subsets can still capture similar importance distinctions as a more detailed partitioning into ten subsets. The correlation increases when comparing $K$=10 and

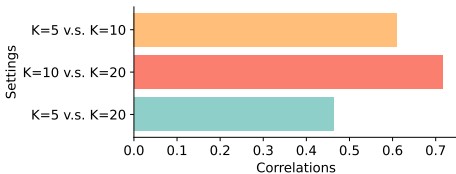

Figure 5: Correlation coefficients between IMAST results for different choices of $K$.

$K$=20. This stronger correlation might be attributed to the fact that splitting into ten or twenty subsets both provides a detailed view of the importance scores, capturing subtler nuances in the model's behavior. On the other hand, there is a weaker correlation between the results of $K$=5 and $K$=20, due to the stark difference in granularity between these scenarios. The coarser partitioning with $K$=5 can miss some finer distinctions in importance scores that are captured when using a larger $K$=20, leading to divergent results. These results demonstrate that while the granularity of subset division (as determined by $K$) can play a role in the final evaluation, the fundamental principles provided by our IMAST remain consistent. Consequently, we observe a trade-off: higher values of $K$ provide more detailed insights at the cost of increased computational demand. Hence, selecting an appropriate $K$ is crucial for balancing the need for precise comparison and computational efficiency.

We further evaluate the comprehensive results of all methods' performance under different choices of $K$. Table 2 presents the averaged performance of interpretation methods across three Transformer models, evaluated by our IMAST with varying choices of $K$. It can be observed that with an increase in $K$, most methods exhibit a slight decline in their IMAST scores. This trend is more pronounced for methods such as Partial LRP (Voita et al., 2019), Transformer Attribution (Chefer et al., 2021b), and Transformer-MM (Chefer et al., 2021a), suggesting that these methods might struggle to maintain faithfulness under a more granular subset division. Conversely, methods like Integrated Gradients (I. G.) and ATTCAT show relatively consistent performance across different $K$ values, indicating a certain level of robustness to the granularity of the subset division. These results reiterate the significance of choosing an appropriate $K$ in the IMAST metric. The optimal value of $K$ depends on the specific needs of the experiment, striking a balance between computational demands and the precision of the faithfulness evaluation.

## C  PROOFS

### C.1  SCALE-INVARIANCE OF IMAST

We will show that our proposed IMAST metric is indeed scale-invariant, as it satisfies

$$\text{IMAST}(aS + b) = \text{IMAST}(S) \qquad (7)$$

for any positive real number $a$ and real number $b$. Here, $S$ denotes the set of importance scores assigned to the input and $aS + b$ denotes a linear transformation performed on each element in set $S$. The IMAST metric is calculated by aggregating the differences between importance scores of

all pairs $(G_i, G_j)$. For each pair, the weight assigned in the calculation is $s(G_i) - s(G_j)$ when $\nabla pred(x, G_i) \geq \nabla pred(x, G_j)$ and $-(s(G_i) - s(G_j))$ otherwise. Consider a linear transformation of the importance scores: $s'(G_i) = as(G_i) + b$. Then, the weight becomes

$$s'(G_i) - s'(G_j) = as(G_i) + b - as(G_j) - b = a(s(G_i) - s(G_j)) \tag{8}$$

for both cases, which implies that the total weight $totalWeight'$ under the transformed scores is

$$totalWeight' = a \times totalWeight. \tag{9}$$

Now consider the faithfulness coefficient $F'$ under the transformed scores. Since the sign of each weight does not change, and the magnitude of each weight is multiplied by $a$, we get

$$F' = aF. \tag{10}$$

Finally, since the IMAST score is calculated as $F'/totalWeight'$, we have

$$\begin{aligned}
\text{IMAST}(aS + b) &= F'/totalWeight' \\
&= aF/a \times totalWeight \\
&= F/totalWeight \\
&= \text{IMAST}(S).
\end{aligned} \tag{11}$$

Hence, the IMAST metric is indeed scale-invariant. It does not depend on the scale of importance scores, which makes the evaluation robust to post-processing steps like normalization or re-scaling.

Note that our demonstration of scale-invariance does not require $a$ to be positive. However, when $a$ is negative, the direction of importance scores is inverted, which would typically lead to an unreasonable explanation. In this situation, even though IMAST maintains scale-invariance, it can consequently yield a low score due to the impact of negative $a$ on the partitioning of subsets. This is because the subsets' arrangement is intrinsically tied to the rankings of importance scores.

## D ENLARGED GRAPHS

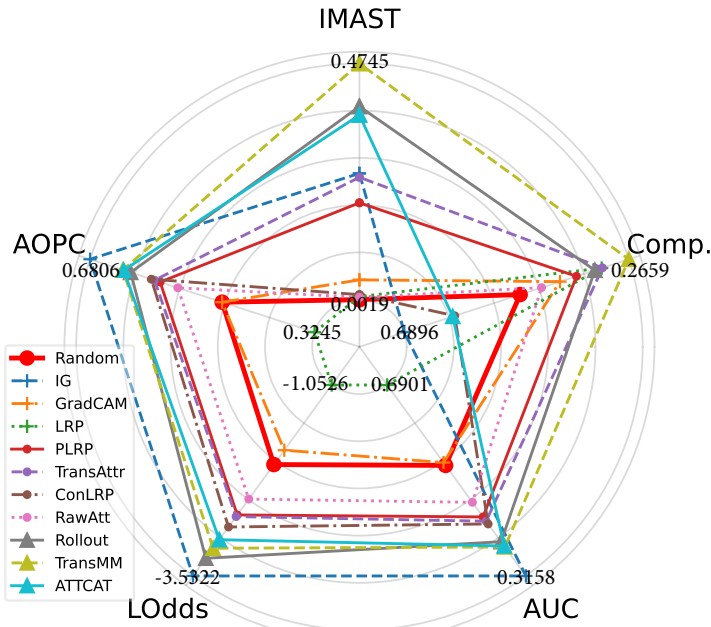

Figure 6: Evaluation results for existing explanation methods as well as Random Attribution, under various metrics. This graph presents results on CIFAR-10 averaged over three Transformer models.

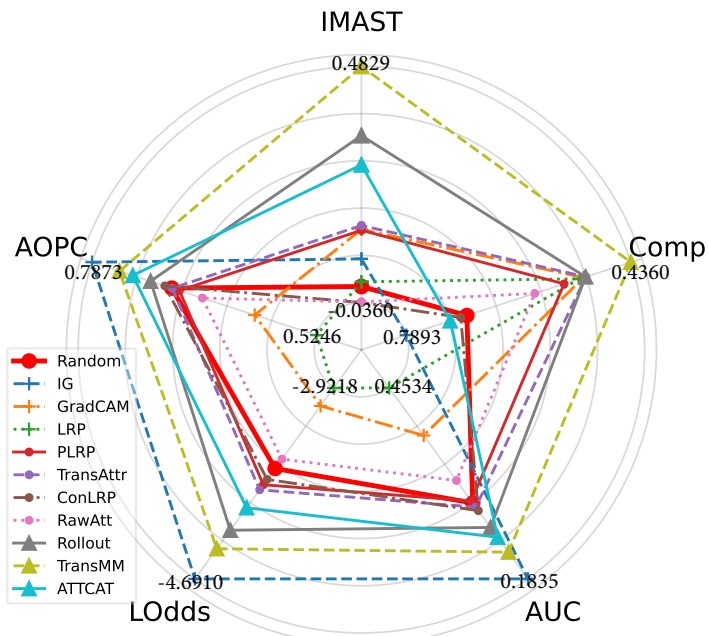

Figure 7: Evaluation results for existing explanation methods as well as Random Attribution, under various metrics. This graph presents results on CIFAR-100 averaged over three Transformer models.

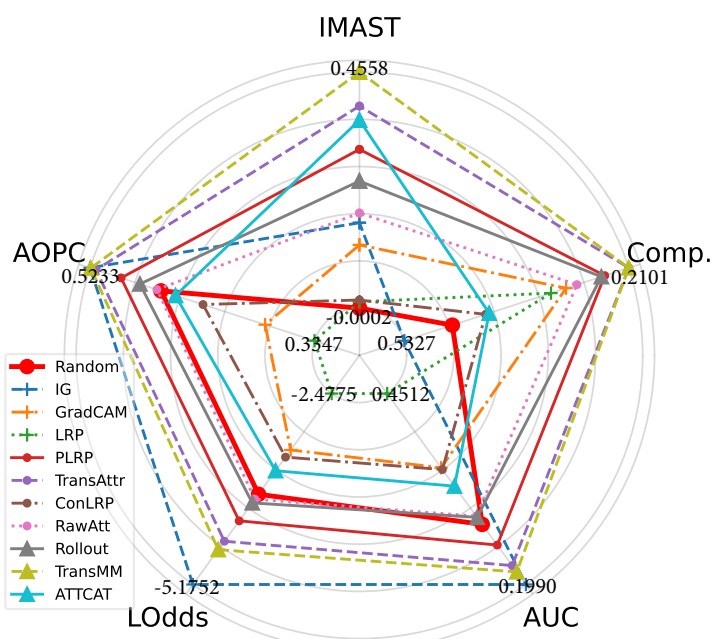

Figure 8: Evaluation results for existing explanation methods as well as Random Attribution, under various metrics. This graph presents results on ImageNet averaged over three Transformer models.

