# OpenReview forum: "IMAST: Importance-Aware Statistical Test for Transformer Interpretability Evaluation"
_ICLR.cc/2024/Conference — ICLR 2024 Conference Withdrawn Submission_

### Official Review · Reviewer_a94J · 2023-10-31

**Soundness:** 2 fair
**Presentation:** 2 fair
**Contribution:** 1 poor
**Rating:** 5
**Confidence:** 5

**Summary:**

This paper is addressing the challenge of faithfulness in explanation methods i.e., how is the explanation score reflected on its impact on final prediction.

**Strengths:**

+ Very important problem
+ New metric for measuring trustworthiness of explanation

**Weaknesses:**

- Would have expected experimentation with more datasets
- Would have expected comparison with other metrics

**Questions:**

1- Why other faithfulness metrics have not been considered for comparison?

---

### Official Review · Reviewer_wvzY · 2023-11-01

**Soundness:** 3 good
**Presentation:** 3 good
**Contribution:** 3 good
**Rating:** 6
**Confidence:** 4

**Summary:**

This paper introduces a novel evaluation framework, the IMportance-Aware Statistical Test (IMAST), to assess the faithfulness of post-hoc explanations for Transformer models. Specifically, IMAST performs statistical comparisons among individual subsets of pixels, aggregating their importance score differences into a faithfulness coefficient $F$. The experimental results expose the limitations of current metrics in their assumption of faithfulness, while also showcasing IMAST's ability to establish a reliable baseline for evaluating faithfulness and providing a robust explanatory result for transformers.

**Strengths:**

1. This paper demonstrates the limitations of existing evaluation metrics and proposes a more faithful metric IMAST for transformers' explanations.
2. The experiments indicate that IMAST outperforms other metrics in terms of the author's core assumption.
3. In facet of paper writing, the exposition provided by the authors is coherent and the systematically structured.

**Weaknesses:**

Despite the above strengths, there are some weaknesses. The contents are as follows.
1. Since there are two patterns for removing features from transformers (removing tokens and pixels ablation), the approach ($Rp(x, G_i)$) used in this paper is not explicitly mentioned. It needs to be described in detail.
2. There is a lack of discussion about the limitations of IMAST.
3. If the paper aims to evaluate the explanation of ViT, the authors should explicitly state this focus. However, if the scope extends beyond ViT, it is necessary to include additional experiments exploring the interpretability of transformers in the field of NLP.
4. Prior to using IMAST and existing metrics, it is necessary for the authors to give more details about the raw evaluation results to support the rank correlations.
5. Regarding the annotation of Figure 3, there is a typo in the quotation mark of `linnet'.
6. Figure 4 shows the evaluation results for all explanation methods. However, it can be observed that the results of IMAST also display an inconsistency among the three datasets. It would be great to add more discussion about this phenomenon.
7. The paper claims that ViT-B, ViT-L, and DeiT-B are adopted, but it remains unclear from the experiment which specific model is employed. Or rather, the results are derived from combining the outputs of the three models. Please provide more explanation.

**Questions:**

Please refer to  "Weaknesses".

---

### Official Review · Reviewer_M9vu · 2023-11-03

**Soundness:** 3 good
**Presentation:** 3 good
**Contribution:** 2 fair
**Rating:** 3
**Confidence:** 4

**Summary:**

The authors propose a new variant of the pixel perturbation approach for quantitatively evaluating importance attribution methods.

In detail, in contrast to evaluating 'pixel-deletion-curves' for which an increasing amount of pixels (sorted by their assigned importance value) is replaced by a baseline value, the authors propose to compare the influence of _groups of pixels_ (grouped by percentile 'bin') on the models' outputs without _cumulatively_ deleting them (the impact of the groups is compared independently).

The authors show that the proposed method better differentiates between random attributions and more advanced importance attribution techniques than other metrics. Further, they show that explanation methods for transformers that aggregate information of multiple network layers as well as gradient information perform better on the proposed metric.

**Strengths:**

The paper has the following strengths:

- The fact that the proposed method is able to provably detect random attributions on average is promising.
- The underlying assumption that groups of more important pixels should have a higher impact on the model prediction than less important ones is intuitive. By avoiding the cumulative deletion, the method introduces less severe artefacts in the input image and allows for directly comparing the groups, which is a good feature of the proposed method.
- The paper is well written and the authors' argumentation is easy to follow.

**Weaknesses:**

Currently, I have the following reservations, which make me hesitant to recommend the acceptance of this submission.

- W1: The authors present the proposed approach (IMAST) as a novel perturbation metric for evaluating explanation methods. However, IMAST seems to be a variation of the Sensitivity-n metric proposed by Ancona et al. (ICLR, 2018). Specifically, Sensitivity-n measures whether the sum of importance values for (randomly sampled) groups of n pixels equals the logit difference after deleting that group of n pixels. Instead of randomly sampling the group of n pixels, the authors of this submission group the pixels by their importance scores and derive a different (but conceptually related) score from the models' behaviour under the deletion of the groups. As such, this seems to be highly relevant work that should be cited and for which a comparative discussion should be added. What is the advantage of the proposed method over Sensitivity-n? How does it complement it?
- W2: The definition of faithfulness (core assumption on page 1) is very narrow and very much tailored to the authors' metric. Faithfulness seems to be too strong of a statement in my opinion for the following reasons. (1) The models' gradient is inherently model-faithful. By definition, it indicates the direction of strongest increase of the output, which is an intrinsic property of the model (and is thus 'faithful' to the model). The fact that the gradient does not perform well on this metric does not change this fact and a different wording should be used (e.g., the behaviour described in the core assumption on page 1 could be a framed as a desideratum for a good/intuitive/human-interpretable explanation, as this is how a human might interpret such heatmaps). (2) The proposed metric (similar to all other input-perturbation-based metrics) conflates the behaviour of the explanation method with that of the model. DNNs are highly non-linear functions, but the proposed metric assumes a highly linear behaviour under the deletion of potentially large groups of input features. How pixel deletions influence the model prediction is not clear a priori (and what faithfulness means then therefore unclear). By not deleting more than K pixels at a time, this metric might be more robust to this issue. A discussion around this would help strengthen the submission in my opinion. E.g., the proposed method could be useful for selecting a model+explanation combination that exhibits the desired property (i.e., core assumption 1).
- W3: The authors frame their work around transformers. However, I fail to see how this metric is transformer-specific in any way. It would significantly strengthen the submission if the discussion of the proposed metric were to be framed in a wider context and if different models were evaluated. As this could help understand differences w.r.t. the applicability of specific explanation methods to certain model types, this could help showcase how the proposed metric yields tangible insights. See also W5.
- W4: The formulation of the metric itself is rather ad-hoc (binary decision whether inequality is violated, penalty/reward given by difference between importance scores). Given that the metric is the central contribution of the submission, a more detailed discussion on pros and cons of the proposed approach as well as alternative metrics would strengthen the paper.
- W5: The main results and the discussion of them are limited. Apart from showing that IMAST yields on average a score of zero for random attributions (Fig. 3), the authors mainly rely on Fig. 4 to argue for the superiority of their method. However, it is not fully clear to me what to conclude from these results, as there is no clear baseline and no clear desired outcome against which to compare the methods.
- W6 (minor): The authors present the results in Table 1 (gradients and aggregating information across layers improve attribution performance) as a new finding based on their method. However, this essentially reflects the core motivation and finding of (Chefer et al., 2021b). Instead of framing this a new finding, I would find it more convincing if this were to be discussed as an indication that IMAST indeed seems to capture some aspect of faithfulness, as it should be expected that only full model explanations (i.e., not only the last layer or only the attention layers) can truly describe the full models (an argument reflected in (Bastings et al., 2020) or (Chefer et al. 2021b)).
- W7 (minor): The authors argue that "previous assessments have demonstrated that attention-based methods generally outperform others [...]" (Sec. 5.4, first paragraph). A citation for this statement is lacking.


Ancona et al. (ICLR, 2018): TOWARDS BETTER UNDERSTANDING OF GRADIENT-BASED ATTRIBUTION METHODS FOR DEEP NEURAL NETWORKS

Bastings et al. (ACL, 2020): The elephant in the interpretability room: Why use attention as explanation when we have saliency methods?

**Questions:**

Please see weaknesses

---

### Official Review · Reviewer_cy4w · 2023-11-04

**Soundness:** 2 fair
**Presentation:** 3 good
**Contribution:** 3 good
**Rating:** 3
**Confidence:** 4

**Summary:**

The authors present an approach to measure the faithfulness of attribution measures. To do this they divide an image into K regions according to ordered attribution score. Then they compute an AUC-type measure (when viewed as expecation between positive and negative labeled pairs) by comparing pairs of regions with positively different impact under masking, and measure the difference of attribution scores over these regions.

**Strengths:**

They propose a new type of attribution measure.
They evaluate it on a larger number of transformer attribution methods.
They analyse its properties to some degree.

**Weaknesses:**

Reasons for leaning to reject are

a the absoluteness of claims, which makes them questionable at least,
b the likely biased implementation of the baselines which rely on ordered sets of scores and cumulative measures,
c and the misleading title.

The first point and the third point are issues with scientific writing quality rather than the proposed method.

c The title is misleading: IMAST performs no statistical testing. It computes a statistical measure. It possibly gives the wrong idea to people who browse titles.

a:
However, we find that current studies tend to neglect the values of im-
portance scores and the model’s confidence in its original predictions

... , and
overlook the values of importance scores in their evaluations.

Sec 5.1 "This implies that they actually evalu-
ate overlapping aspects of interpretations (mainly the effect of progressive feature removal) while
overlooking the faithfulness assumption."

The claim the AUC/AOPC does not consider faithfulness as an absolute statement is false (unless one would forget to order the scores). This is not the same as a claim that the proposed measure considers faithfulness to a better degree.

Even Fig 3 does not support the claim of the absolute statement. AOPC changes, not on a nice scale as IMAST, but it does. Making dubious statements due to absoluteness of claims is a reason to reject a paper.

a separate point:
Fig 3 - random attribution - The first set has the highest cumulative score, and the difference between cumulative sets is too large and too linear to come from random attribution. While there can be random fluctuations, the regular linear decrease is very unlikely to occur under a random attribution.

It is random but it appears rather to favor the important object - as also evidenced by the fact that removing 20% of the content removes 80% of the confidence score. Since scores are accumulated over many pixels, this is unlikely to occur in a random draw.

This is not proof but an indicator that something might have been implemented not well or wrongly for the evaluation of the cumulative type scores.

That said, "such Random Attribution represents a complete absence of faithfulness"
-- holds only in expectation. For a particular draw, it can have a partial correlation.


b: The baseline comparison to cumulative perturbation methods has a flaw:
AOPC needs to be measured at the scale of relevant evidence fraction, not in fixed 10% steps. Reason being:

Usually, evidence for discriminatively trained methods is very sparse, such as within the top-5% or worst case top-20% of the total area. Thus steps should be taken on a scale which progresses comparatively to the amount of high scoring regions, for example using 0.5% of image area until 10 % or 20%.

See for an example the percentage scale in Fig 4 in
https://arxiv.org/pdf/2202.07304.pdf .

b continued:
The other point of criticism: it is not clear how the regions of 10% size are chosen. For images with spatial localization one would not choose disjoint sets of pixels all across the image for AOPC and the others - this ignores the local correlation structure of image data, but instead choose iteratively from regions computed using overlapping sliding windows.
This makes it hard, however to make a clear cut in 10% blocks after one has removed a few blocks.


minor issues:

From the perspective of the core
assumption, such Random Attribution represents a complete absence of faithfulness, as the assigned
importance scores bear no relationship to the actual impact of each input element. As a result, an
ideal metric for assessing faithfulness should return a distinct score when applied to Random Attri-
bution, essentially setting a baseline indicative of a complete lack of meaningful understanding of
the model’s prediction.

This is true only in expectation, not for a particular random draw. A particular draw may have a correlation to some regions of importance. Again the issue here is with absolute statements.



As displayed, the correlation scores between our IMAST and other existing metrics range from 0.18
to 0.22 (in this analysis, a result of 1 indicates a complete correlation, while a result of 0 indicates
no correlation). These low scores signify minimal congruence in their evaluation, indicating that
existing metrics may not effectively capture the adherence of an explanation to the central assump-
tion of faithfulness.

This is a statement which assumes that IMAST is the ground truth. This contains a logical fallacy in the argumentation. This is to be shown, not to be assumed.


A phenomenon across three
datasets in our experiment exemplifies this problem: while I.G. performs best on AOPC, LOdds,
and AUC (with Most Relevant First removal), it is surprisingly the worst on Comprehensiveness
(with the reverse removal order). This indicates that current metrics are significantly inconsistent
w.r.t. hyperparameters such as removal orders (Rong et al., 2022).

The reviewer disagrees here. Most Relevant First and Least Relevant first measure different aspects of an attribution - identification of most important region, versus attributions for least important regions as a measure of noisyness of the attribution. So the difference is on purpose.

IMAST puts all in one measure, whereas the above measure two ends of it.

The reviewer believes that IMAST provides a publishable contribution, however it needs a more appropriate title, less absolute claims, and a more precise comparison against the baselines (running them in different, smaller, percentages, with spatially coherent regions ).

The reviewer is somewhere between a reject and a marginally below for this paper.
Once the issues are addressed, the reviewer is willing to reconsider the given score.

**Questions:**

Specifically, following the literature con-
vention, we divide the image into ten disjoint pixel subsets

How these pixel subsets are actually chosen ?

Can the authors publish their used code to evaluate cumulative type measures?

On how many images Figure 4 was actually computed?